# Resource partitioning among brachiopods and bivalves at ancient hydrocarbon seeps: A hypothesis

Steffen Kiel[1], Jörn Peckmann[2]*

**1** Swedish Museum of Natural History, Department of Palaeobiology, Stockholm, Sweden, **2** Universität Hamburg, Center for Earth System Research and Sustainability, Institute for Geology, Hamburg, Germany

* joern.peckmann@uni-hamburg.de

**Data Availability Statement:** All relevant data are in the paper.

**Funding:** The authors received no specific funding for this work.

## Abstract

Brachiopods were thought to have dominated deep-sea hydrothermal vents and hydrocarbon seeps for most of the Paleozoic and Mesozoic, and were believed to have been outcompeted and replaced by chemosymbiotic bivalves during the Late Cretaceous. But recent findings of bivalve-rich seep deposits of Paleozoic and Mesozoic age have questioned this paradigm. By tabulating the generic diversity of the dominant brachiopod and bivalve clades–dimerelloid brachiopods and chemosymbiotic bivalves–from hydrocarbon seeps through the Phanerozoic, we show that their evolutionary trajectories are largely unrelated to one another, indicating that they have not been competing for the same resources. We hypothesize that the dimerelloid brachiopods generally preferred seeps with abundant hydrocarbons in the bottom waters above the seep, such as oil seeps or methane seeps with diffusive seepage, whereas seeps with strong, advective fluid flow and hence abundant hydrogen sulfide were less favorable for them. At methane seeps typified by diffusive seepage and oil seeps, oxidation of hydrocarbons in the bottom water by chemotrophic bacteria enhances the growth of bacterioplankton, on which the brachiopods could have filter fed. Whereas chemosymbiotic bivalves mostly relied on sulfide-oxidizing symbionts for nutrition, for the brachiopods aerobic bacterial oxidation of methane and other hydrocarbons played a more prominent role. The availability of geofuels (i.e. the reduced chemical compounds used in chemosynthesis such as hydrogen sulfide, methane, and other hydrocarbons) at seeps is mostly governed by fluid flow rates, geological setting, and marine sulfate concentrations. Thus rather than competition, we suggest that geofuel type and availability controlled the distribution of brachiopods and bivalves at hydrocarbon seeps through the Phanerozoic.

## Introduction

The idea of bivalves replacing brachiopods as the dominant benthic filter feeders over the course of the Phanerozoic is one of the oldest macroevolutionary patterns discussed in paleontology [1–3]. Originally observed in the rich fossil record of shallow marine environments, a

**Competing interests:** The authors have declared that no competing interests exist.

similar pattern was also seen at deep-sea hydrothermal vents and hydrocarbon seeps [4]. These ecosystems differ radically from all others in being based on chemosynthetic primary production rather than photosynthesis [5]. The evolution of the chemosynthesis-based faunal communities may therefore be buffered from mass extinctions and other disruptions of photosynthesis-based food chains [6–8] and may instead be driven by events affecting the discharge of the reduced chemicals (referred to as geofuels hereafter) that fuel the chemosynthesis-based food chain [9]. The animals that dominate chemosynthesis-based ecosystems show extensive physiological adaptations, commonly involving a symbiosis with chemotrophic bacteria, resulting in faunal communities with a low diversity but high abundance of highly specialized animals [10, 11].

A first compilation of Phanerozoic vent and seep sites with brachiopods and/or bivalves indicated a pattern of a Paleozoic to middle Mesozoic dominance of brachiopods in these ecosystems, and the chemosymbiotic bivalves became dominant from the Late Cretaceous onward [4]. Subsequent research confirmed a number of Paleozoic and early Mesozoic seep deposits dominated by brachiopods, including *Septatrypa* in the Silurian [12], *Dzieduszyckia* in the Devonian [13], *Ibergirhynchia* in the Carboniferous [14], *Halorella* in the Triassic [15], and *Sulcirostra* and *Anarhynchia* in the Jurassic [16–18], supporting this hypothesis. However, also discovered were seep deposits at which inferred chemosymbiotic bivalves were a major faunal element, including the modiomorphid genus *Ataviaconcha* at Silurian and Devonian sites in Morocco [19, 20] and kalenterid and anomalodesmatan genera at Triassic sites in Turkey [21, 22]. These findings challenge the claim of predominantly brachiopod-dominated pre-Cretaceous vents and seeps, and raise the questions why some sites were dominated by brachiopods and others by bivalves.

A further complication in this context is that the feeding strategy (or strategies) of vent and seep-inhabiting brachiopods is essentially unknown. The large size of certain species and brachiopod dominance at some sites may intuitively suggest that they were chemosymbiotic [18, 23, 24]. However, brachiopods are virtually absent from extant vents and seeps, and the few known examples are filter feeder that take advantage of the hard substrate provided by authigenic carbonates exposed at some seeps [25]. Furthermore, the general brachiopod bauplan lacks certain features, such as gills and a closed cardio-vascular system, which are important for hosting chemosymbionts in bivalves or tube worms [26, 27] and make brachiopods ill-suited for coping with the toxicity of hydrogen sulfide. Furthermore, the brachiopods that formed mass occurrences at ancient seeps are not drawn randomly from the brachiopod tree of life but instead belong, with one exception, to a single clade: the Dimerelloidea [28]. The one exception is the genus *Septatrypa* from a Silurian seep deposit in Morocco [12], which belongs to a different rhynchonellate order than the dimerelloids. Because insights into the feeding strategy of seep-dominating brachiopods are only available for the dimerelloids (see below), our study focuses exclusively on members of this clade. One intriguing feature of many seep deposits dominated by dimerelloids is the sheer abundance of the brachiopods, which by far exceeds the abundance of chemosymbiotic bivalves at fossil seep deposits ([13, 15, 17, 29], own observations).

Here we present the hypothesis that dimerelloid brachiopods and chemosymbiotic bivalves coexisted at hydrocarbon seeps during the Paleozoic and Mesozoic by partitioning the locally available geofuels. We propose that the presence, absence, or relative abundance of each clade at a given site was largely controlled by the chemical composition of the seep fluids (the proportions of sulfide, methane, and/or oil), which in turn was influenced by seepage intensity and perhaps seawater sulfate concentrations. Our hypothesis is based on (i) a tabulation of the diversity of the ecologically dominant bivalve and brachiopod genera at seeps through the Phanerozoic; (ii) recent improvements in geochemically assessing the composition of fluids

and the intensity of fluid flow at ancient seeps; and (iii) a set of derivations and assumptions on the paleoecology of the dominant brachiopod and bivalve clades at ancient hydrocarbon seeps.

## Approach

### Compilation of generic diversity of bivalves and brachiopods

Modern seep communities are characterized by the low diversity but high abundance of a few taxa that are able to take advantage of the unique food resources at seeps [5, 30]. Thus our compilation of brachiopod and bivalve diversity at seeps includes only the ecologically dominant clades instead of the full range of genera known from fossil seep deposits to avoid the results being blurred by chance occurrences or by local taxa fortuitously taking advantage of the abundance of food at a seep site (known as 'vagrants' or 'background taxa', cf. [31] Sibuet and Olu, 1998). Among bivalves, only chemosymbiotic or in the case of extinct taxa, inferred chemosymbiotic taxa, were included. Although chemosymbiotic bivalves may occasionally be rare, in general they dominate seep deposits numerically (cf. [9] Kiel, 2015; [32] Campbell, 2006).

Among brachiopods, only dimerelloid genera reported from geochemically confirmed seep deposits are included because (i) with a single exception (see below), only dimerelloids occurred at ancient seeps in rock-forming quantities; all other brachiopods reported from ancient seeps (including various terebratulids, i.e. [33–37]) represent minor faunal elements that most likely took advantage of exposed hard substrate [28], and (ii) the feeding strategies of brachiopods at ancient seeps remain unclear except for the dimerelloids, for which some clues are available [38]. The only exception to (i) is *Septatrypa*, which occurs in rock-forming quantities in a Silurian seep deposit from Morocco [12]. This genus belongs to a different rhynchonellate order than the dimerelloids, hence we refrain from extending the feeding strategy inferred from the Cretaceous dimerelloid *Peregrinella* to this Silurian genus.

The dataset includes 42 bivalve and seven brachiopod genera; their stratigraphic distributions, life habits (epifaunal, semi-infaunal, and infaunal), and all relevant references are shown in Table 1. To assess a potential sampling bias, we also compiled the number of seep-bearing rock units at which these taxa were found (Table 2), as done in a previous quantitative study on seep faunas [8].

### Proxies for fluid chemistry and flow intensity at ancient seeps

Criteria used to reconstruct the composition of seep fluids and seepage intensity are based on the mineralogy and microfabric of authigenic carbonate and sulfide minerals, stable isotope signatures of authigenic minerals, and lipid biomarkers [133–140]. This set of methods, however, does not allow to reliably discern methane-seep and oil-seep deposits. The use of lipid biomarkers seems an obvious approach for such discrimination, but is hampered by the facts that (i) sulfate-driven anaerobic oxidation of methane occurs at oil seeps too [141] and (ii) the prokaryotes responsible for anaerobic degradation of oil components in marine settings (i.e. sulfate-reducing bacteria; [142]) may yield similar biomarkers like the sulfate-reducing bacteria involved in anaerobic oxidation of methane. Even more problematic, the great abundance of oil components in some seep deposits tends to mask the lipid biomarkers reflecting local biogeochemical processes [143]. Such masking by oil-derived components is a particular problem for the recognition of possible ancient oil-seep deposits, since the timing of oil ingress (syngenetic vs. epigenetic) is commonly difficult to constrain [13]. The sheer presence of pyrobitumen (i.e. metamorphosed oil) in ancient seep limestones is consequently not sufficient proof for oil seepage. These problems prompted the development of an inorganic geochemical

**Table 1. Dimerelloid brachiopod genera and (inferred) chemosymbiotic bivalve genera in ancient hydrocarbon-seep deposits.** New brachiopod and bivalve genera established since Mike Sandy's review of dimerelloid brachiopods as seep-inhabitants in 1995 [23] are marked by an asterisk (*); see section 'Diversity pattern' for reasoning.

| | |
|---|---|
| **NEOGENE** | |
| Dimerelloid brachiopods: | none. |
| Infaunal bivalves: | *Acharax* [39], *Anodontia* [40], *Channelaxinus** [41], *Cubatea** [42], *Elliptiolucina**, *Elongatolucina** [43], *Isorropodon* [44], *Lucinoma, Meganodontia** [41], *Megaxinus* [45], *Myrteopsis, Nipponothracia**, Pegaphysema* [43], *Pliocardia* [46], *Solemya* [47], *Thyasira* [48]. |
| Semi-infaunal bivalves: | *Archivesica* [41], *Callogonia* [49], *Calyptogena* [50], *Conchocele* [42, 47], *Gigantidas** [51], *Notocalyptogena** [52], *Pleurophopsis* (= *Adulomya*) [42, 53, 54]. |
| Epifaunal bivalves: | *Bathymodiolus* [41, 51]. |
| **PALEOGENE** | |
| Dimerelloid brachiopods: | none. |
| Infaunal bivalves: | *Acharax* [42, 55], *Amanocina**, *Cubatea**, *Elliptiolucina**, *Elongatolucina** [43, 56], *Epilucina* [57], *Lucinoma* [58], *Maorityas* [59], *Nipponothracia** [43], *Nucinella* [55], *Nymphalucina* [43], *Pliocardia* [42], *Rhacothyas** [33], *Solemya* [60], *Thyasira* [59]. |
| Semi-infaunal bivalves: | *Conchocele* [59], *Hubertschenckia* [44], *Pleurophopsis* [42]. |
| Epifaunal bivalves: | *Bathymodiolus, Idas, Vulcanidas** [60]. |
| **LATE CRETACEOUS** | |
| Dimerelloid brachiopods: | none. |
| Infaunal bivalves: | *Acharax* [61], *Amanocina** [43], *Cubatea** [43], *Miltha* [61], *Myrtea* [61], *Nucinella* [62], *Nymphalucina* [43], *Solemya* [63, 64], *Tehamatea** [43], *Thyasira* [48, 59]. |
| Semi-infaunal bivalves: | *Caspiconcha** [65, 66], *Conchocele* [59, 63]. |
| Epifaunal bivalves: | none. |
| **EARLY CRETACEOUS** | |
| Dimerelloid brachiopods: | *Peregrinella* [29, 38]. |
| Infaunal bivalves: | *Acharax* [67], *Amanocina** [43], *Cretaxinus** [68], *Cubatea** [43], *Nucinella* [67–69], *Solemya* [64, 68, 70], *Tehamatea** [43], *Thyasira* [59]. |
| Semi-infaunal bivalves: | *Caspiconcha** [65]. |
| Epifaunal bivalves: | none. |
| **JURASSIC** | |
| Dimerelloid brachiopods: | *Anarhynchia* [18], *Cooperrhynchia* [71], *Sulcirostra* [17]. |
| Infaunal bivalves: | *Acharax* [67], *Beauvoisina** [43], *Nucinella* [68], *Solemya* [68], *Tehamatea** [37]. |
| Semi-infaunal bivalves: | *Caspiconcha** [65]. |
| Epifaunal bivalves: | none. |
| **TRIASSIC** | |
| Dimerelloid brachiopods: | *Halorella* [15, 21]. |
| Infaunal bivalves: | *Nucinella* [15], *Aksumya** [22]. |
| Semi-infaunal bivalves: | *Terzileria**, *Kasimlara** [22]. |
| Epifaunal bivalves: | none. |
| **CARBONIFEROUS** | |

*(Continued)*

**Table 1.** (Continued)

| | |
|---|---|
| Dimerelloid brachiopods: | *Ibergirhynchia** [72]. |
| Infaunal bivalves: | 'solemyid' [14]. |
| Semi-infaunal bivalves: | none. |
| Epifaunal bivalves: | none. |
| **DEVONIAN** | |
| Dimerelloid brachiopods: | *Dzieduszyckia* [13]. |
| Infaunal bivalves: | *Dystactella* [20]. |
| Semi-infaunal bivalves: | *Ataviaconcha** [20]. |
| Epifaunal bivalves: | none. |

**Table 2. Seep-bearing rock units or equivalents, sorted into the same geologic time bins as the genera in Table 1; Fm = Formation.**

| Site(s) | Rock unit or equivalent | Reference |
|---|---|---|
| **NEOGENE** | | |
| Fukaura town | Akaishi Fm | [50] |
| Ogasawara's slumped block | Aokiyama Fm | [47] |
| Stirone River seep complex | Argille Azzurre Fm | [73] |
| LACM loc. 6132, USGS M2790 | Astoria Fm | [44] |
| Akanuda Limestone | Bessho Fm | [74] |
| Bexhaven, Karikarihuata, Moonlight North, Rocky Knob, Tauwhareparae, Waipiro | Bexhaven Limestone | [51] |
| Liog-Liog Point | Bata Fm | [75] |
| Takangshan quarry | Gutingkeng Fm | [76] |
| Ikegami | Hayama Fm | [77] |
| Shimo-sasahara, near Yatsuo | Higashibessho Fm | [46] |
| Oinomikado & Kanehara 1938 loc. | Higashiyama Oil field | [47] |
| Saitama conglomerate | Hiranita Fm | [78] |
| Nagasawa & Oyamada's 1996 loc. | Hongo Fm | [47] |
| Kamada's Honya loc. | Honya Fm | [47] |
| Cantera Portugalete | Husillo Fm | [42] |
| Haunui, Ugly Hill, Wanstedt | Ihungia Series | [79] |
| slumped blocks, YDFAB 1993 | Ikedo Fm | [47] |
| Joban coal field | Kabeya Fm | [80] |
| Doguchi Bridge | Kawazume Fm | [47] |
| Izura Kanko Hotel | Kokozura Fm | [49] |
| Matsudai, Sugawa | Kurokura Fm | [81] |
| Freeman's Bay, Godineau River, Jordan Hill | Lengua Fm | [42] |
| Casa Cavalmagra, Case Rovereti, Castellvecchio, Le Colline, Montepetra | Marnoso-arenacea Fm | [41] |
| Kanie's 1991 juvenile Calyptogena loc. | Misaki Fm | [47] |
| Morai, Otatsume's 1942 loc. | Morai Fm | [47] |
| loc. M2 of Shikama & Kase, 1976 | Morozaki Group | [47] |
| Limestone nodule in Kochi | Muroto Fm | [82] |
| Ozaki 1958 loc. | Naari Fm | [83] |

(*Continued*)

**Table 2.** (Continued)

| Site(s) | Rock unit or equivalent | Reference |
|---|---|---|
| Nadachi Signal Station | Nadachi Fm | [84] |
| Amano et al 1994 loc. | Nanbayama Fm | [47] |
| Nakanomata seep deposit | Nodani Fm | [85] |
| Rekifune seep | Nupinai Fm | [86] |
| Kanno & Akatsu 1972 loc. | Nupinaigawa mudstone | [47] |
| Yokohama City | Ofuna Fm | [47] |
| Kita-Kuroiwa | Ogaya Fm | [74] |
| Sakurai's 2003 loc. | Ogikubo Fm | [47] |
| Hayashi's 1973 localities | Ohno Fm | [47] |
| Ogasawara 1986 Akita loc. | Onnagawa Fm | [47] |
| Huso Clay Member | Pozon Fm | [87] |
| Quinault seep | Quinault Fm | [88] |
| Hayashi & Miura's 1973 loc. | Ryusenji Fm | [47] |
| Buton asphalt deposit | Sampalokossa Beds | [89] |
| Matsumoto & Hirata's 1972 Shizuoka loc. | Setogawa Group | [82] |
| Oshima | Shikiya Fm | [90] |
| Kawaguchi/Kotto | Shiramazu Fm | [91] |
| SOFZ—Baths Cliffs fauna | SOFZ | [87] |
| Joban coal field | Taira Fm | [80] |
| Kanno & Ogawa 1964 loc. | Takinoue Fm | [92] |
| Kanno's 1967 Tokyo loc. | Tateya Fm | [47] |
| Oinomikado & Kanehara's 1938 loc. | Teradomari Fm | [47] |
| Sasso delle Streghe | Termina Fm | [41] |
| Abisso Mornig, Casa Carnè, Casa Piantè | Tossignano marls | [41] |
| Tanaka's Matsumoto City loc. | Uchimura Fm | [47] |
| Katto & Masuda's 1978 pyrite loc. | Uematsu Fm | [47] |
| Matsumoto's 1966 & 1971 Shizuoka locs. | Wappazawa Fm | [47] |
| Kanehara's 1937 loc. | Yunagaya and Shirado Groups | [93] |
| **PALEOGENE** | | |
| Fossildalen | Basilika Fm | [35] |
| Buje petrol station | Central Istria flysch | [94] |
| Diapiric mélange, Joes River | Diapiric melange | [87] |
| Angela Elmira asphalt mine | Elmira Asphalt | [95] |
| Bear River (LACMIP loc. 5802) | equivalent of Lincoln Creek Fm | [96] |
| Belen | Heath shales | [95] |
| LACMIP loc. 12385, CSUN loc. 1583 | Humptulips Fm | [96] |
| LACMIP loc. 17101 | Jansen Creek Member | [60] |
| Rock Creek Oregon, Vernonia-Timber Road | Keasey Fm | [97] |
| CR2, UWBM loc. B-7451, LACMIP loc. 5843, LACMIP loc. 16504, SR1-SR4 | Lincoln Creek Fm | [60, 98–100] |
| Bullman Creek, LACMIP loc. 6958, Shipwreck Point | Makah Fm | [58, 59, 101] |
| Cima Sandstone lentil | Moreno Fm | [102] |
| Kami-Atsunai railway station | Nuibetsu Fm | [103] |
| Urahoro concretion | Oomagari Fm | [47] |
| Palmar-Molinera-Road | Palmar-Molinera | [95] |
| *Huberschenkia*-loc. (Yayoi site) | Poronai Fm | [104] |

(*Continued*)

**Table 2.** (Continued)

| Site(s) | Rock unit or equivalent | Reference |
|---|---|---|
| East Twin River, LACMIP locs. 15621, 6295, Whiskey Creek | Pysht Fm | [59, 100, 101, 105] |
| North Slope | Sagavanirktok Fm | [32] |
| Kiritachi | Sakasagawa Fm | [59] |
| Tanami | Shimotsuyu Fm | [55] |
| Columbia River, UWBM loc. B-7446 | Siltstone of Shoalwater Bay | [60] |
| West Fork of Grays River | Siltstone of Unit B | [59] |
| Lomitos | Talara Fm | [95] |
| Wagonwheel seep CSUN loc. 1580 | Wagonwheel Fm | [57] |
| **LATE CRETACEOUS** | | |
| Awanui GS 688, Waipiro I, Waipiro III | East Coast Allochthon | [69] |
| Guenoc Ranch | Great Valley Group | [64] |
| Maeshima | Himenoura Group | [59] |
| Seymour Island | Lopez de Bertodano Fm | [63] |
| Romero Creek | Moreno Fm | [64] |
| Sada Limestone | Nakamura Fm | [106] |
| Omagari lens, Yasukawa seep | Omagari Fm | [107, 108] |
| Tepee Buttes | Pierre Shale | [109] |
| Snow Hill Island | Snow Hill Island Fm | [63] |
| Alton Sink, North & South Cottonwood Wash | Tropic Shale | [110] |
| Obira-cho | Yezo Supergroup | [61] |
| **EARLY CRETACEOUS** | | |
| Sassenfjorden carbonates | Agardhfjellet Fm | [111] |
| near Freiberg | Beskidy Range | [112] |
| Eagle Creek | Budden Canyon Fm | [65] |
| Yongzhu bridge | Chebo Fm | [113] |
| Bonanza Creek | Chisana Fm | [114] |
| Prince Patric & Ellef Ringnes isl&s | Christopher Fm | [115] |
| Bear Creek, Foley Canyon, Rocky Creek | Crack Canyon Fm | [64] |
| Awanui I & II | East Coast Allochthon | [69] |
| Novaya Zemlya III sandy limestone | float | [116] |
| Little Indian Valley | Fransiscan Complex | [64] |
| Gravelly Flat | Gravely Flat Fm | [64] |
| East Berryessa, Knoxville, Rice Valley, West Berryessa, Wilbur Springs | Great Valley Group | [64] |
| Baska | Hradiště Fm | [117] |
| Cold Fork of Cottonwood Creek | Lodoga Fm | [118] |
| Musenalp | Musenalp | [119] |
| Ispaster | Ogella unit | [120] |
| W. Kuban | Oubine Valley | [121] |
| Planerskoje | Planerskoje section | [122] |
| East of Lhasa | Sangxiu Fm | [123] |
| Sinaia Beds | Sinaia Fm | [124] |
| Koniakov, Koniakover Schloss, Raciborsko | Upper Grodziszcze beds | [29, 125] |
| Curnier, Rottier | Vocontien Basin | [38] |
| Kuhnpasset Beds | Wollaston Forland | [70] |
| Ponbetsu, Utagoesawa | Yezo Supergroup | [66, 126] |

*(Continued)*

**Table 2.** (Continued)

| Site(s) | Rock unit or equivalent | Reference |
|---|---|---|
| **JURASSIC** | | |
| Sassenfjorden carbonates | Agardhfjellet Fm | [111] |
| Gateway Pass Limestone Bed | Atoll Nunataks Fm | [127] |
| Novaya Zemlya I & II | float | [116] |
| Charlie Valley | Fransiscan Complex | [64] |
| NW Berryessa, Stony Creek | Great Valley Group | [64] |
| Copper Island | Inklin Fm | [18] |
| Seneca | Keller Creek Fm | [17] |
| Paskenta | Stony Creek Fm | [128] |
| Beauvoisin | Terres Noires Fm | [129] |
| **TRIASSIC** | | |
| Terziler and Dumanlı | Kasimlar shales | [21] |
| Graylock Buttes | Rail Cabin mudstone | [15] |
| **CARBONIFEROUS** | | |
| Tentes Mound | Calcaires de l'Iraty | [130] |
| Ganigobis | Ganigobis Shale Member | [131] |
| Iberg seep | Iberg reef | [14] |
| **DEVONIAN** | | |
| Sidi Amar | Devonian-Carboniferous mélange | [13] |
| Hollard Mound | *Pinacites* limestone | [132] |

proxy for oil seepage. Molybdenum to uranium ratios in conjunction with rare earth element contents of seep limestones have been shown to allow oil-seep and methane-seep deposits to be discriminated [144]. The application of this proxy resulted in the confirmation that Late Devonian limestones from Morocco with the dimerelloid brachiopod *Dzieduszyckia* formed at oil seeps. Future work will have to reveal if some of the other seep limestones with dimerelloids, for which the presence of pyrobitumen was documented, are oil-seep deposits as well.

Another crucial aspect for our reconstruction of the adaptation of bivalves and brachiopods to seep ecosystems concerns the mode in which methane was predominantly oxidized (i.e. anaerobic vs. aerobic methanotrophy). It seems straightforward that episodes of low seawater sulfate concentration favor the release of methane into bottom waters [9, 145], with less methane being oxidized at the sulfate-methane transition zone and more methane available for aerobic methane-oxidizing bacteria. Interestingly, also the mode of seepage (i.e. advective vs. diffusive) is likely to affect the relative proportions of anaerobic and aerobic methanotrophy. Although possibly counterintuitive at first glance, it has been shown that more methane tends to permeate the barrier at the sulfate-methane transition zone formed by sulfate-driven anaerobic oxidation of methane at diffusive seeps compared to advective seeps [146–148]. This circumstance agrees with the observation of more abundant biomarkers of aerobic methanotrophic bacteria in seep deposits reflecting diffusive seepage (Natalicchio et al., 2015) and the inferred affinity of *Peregrinella* to diffusive seepage and aerobic methanotrophy [38]. Likewise, seeps with advective flow will tend to be characterized by high concentrations of hydrogen sulfide–resulting from pronounced sulfate-driven anaerobic oxidation of methane at the sulfate-methane transition zone–whereas at seeps with diffusive flow more methane will be oxidized with molecular oxygen by aerobic methanotrophs.

## Assumptions on chemosymbiosis in fossil bivalves

For the sake of our hypothesis, we assume that all bivalve clades that dominated seep deposits before the Cenozoic era were hosting thiotrophic (i.e. sulfide-oxidizing) symbionts only. For most clades, including the Solemyidae, Nucinellidae, Thyasiridae, and Lucinidae, this is a fair assumption based on the actualistic principle: members of these families host thiotrophic symbionts only [149]. The only bivalve clade known to harbor methanotrophic symbionts is the Bathymodiolinae [150, 151]. A detailed study on their evolutionary history [152] showed that the path to methanotrophic symbiosis is difficult: first, only 13 out of 52 investigated species harbor methanotrophs; second, intracellular rather than extracellular symbiont location seems to be required to host methanotrophs; and third, methanotrophic symbiosis was acquired fairly recently in the evolutionary history of the bathymodiolins (in the early Miocene), while the original thiotrophic symbiosis goes much further back in time [152]. Remarkable in this context is that other bivalve families with intracellular symbionts have apparently not developed methanotrophic symbiosis, despite having a similarly long (Vesicomyidae; cf. [153] Kiel, 2010) or much longer evolutionary history (Solemyidae, Lucinidae; cf. [20] Hryniewicz et al., 2017; [154] Taylor et al., 2011).

Inferring chemosymbiosis or even symbiotic types is much harder in extinct taxa such as the modiomorphids–a clade commonly found at ancient vents and seep [20, 155]–because there is presently no way to proof chemosymbiosis in the fossil record. However, some clues may be drawn from the geologic history of the modiomorphid/kalenterid genus *Caspiconcha*, which is found in many Late Jurassic to Late Cretaceous seep deposits around the world [66, 70, 122]. *Caspiconcha* was common during most of the Early Cretaceous but declined in abundance and eventually disappeared after marine sulfate concentrations–and hence sulfide availability at seeps–dropped in the Aptian [9, 66, 145]. If *Caspiconcha* had had methanotrophic symbionts, it should not have been affected by the low sulfate concentrations; on the contrary, it should have thrived due to the higher availability of methane at seeps (see above). But *Caspiconcha* responded to the mid- to Late Cretaceous low sulfate concentrations in a way expected for a taxon with thiotrophic symbionts. Based on this observation, we assume that *Caspiconcha*, and seep-inhabiting modiomorphids/kalenterids during the Phanerozoic in general, had thiotrophic rather than methanotrophic symbionts. Furthermore, because virtually all extant bivalves taking up geofuels for their symbionts from pore water have thiotrophic symbionts [11, 149], the infaunal and semi-infaunal lifestyle of the inferred chemosymbiotic bivalves at pre-Cenozoic seeps suggests that they relied on thiotrophy rather than methanotrophy.

## The resource partitioning hypothesis: Outline and arguments

### Diversity pattern

The bivalve genera at seeps are of low diversity during the Paleozoic followed by a continuous increase in diversity since the Triassic (Fig 1). Prior to the Cenozoic, this increase in diversity is mostly among infaunal genera, plus a few semi-infaunal ones; epifaunal bivalves appeared only in the Cenozoic (Fig 1). The continuous rise in bivalve diversity at seeps, at least since the Mesozoic, appears to mirror the general Phanerozoic increase in bivalve diversity [2]. But the low diversity of semi-infaunal and epifaunal bivalves at seeps and their rapid diversification in the Cenozoic are unlike the general Phanerozoic pattern of bivalve ecospace occupation with its similar proportions of infaunal, semi-infaunal, and epifaunal taxa in the Mesozoic and Cenozoic [156]. This bias toward infaunal taxa might result from our focus on chemosymbiotic bivalves. Indeed, the bivalve diversification pattern at seeps is quite similar to that of the most diverse clade of shallow-water chemosymbiotic bivalves–the Lucinidae [157]–which also

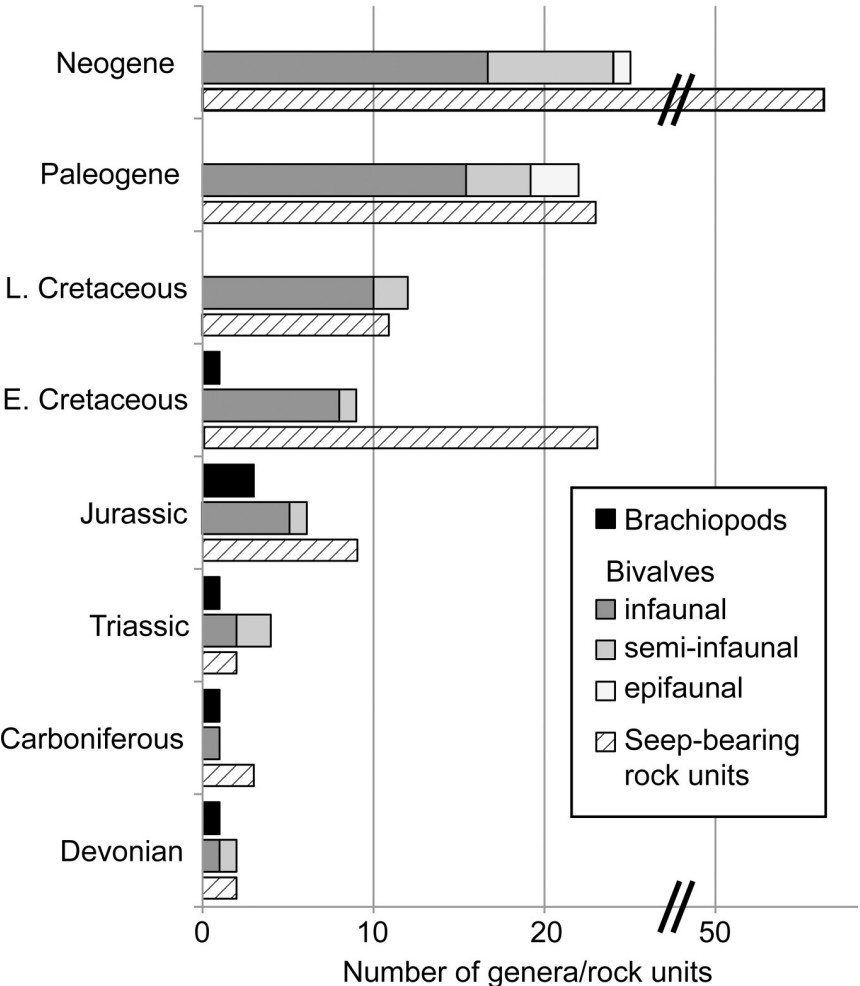

**Fig 1. Phanerozoic generic diversity of chemosymbiotic bivalves and dimerelloid brachiopods at hydrocarbon seeps, and the number of seep-bearing rock units.** Note break in scale and that the Permian was omitted because no confirmed seep deposits have been reported from this period to date. E. = Early, L. = Late.

shows low diversity during the Paleozoic and a continuous increase starting in the Mesozoic [158]. One may thus argue that bivalve diversity at seeps follows the diversity of chemosymbiotic bivalves in shallow water. Epifaunal and semi-infaunal chemosymbiotic bivalves such as bathymodiolins and vesicomyids are virtually absent from shallow water [149] and appear to be a unique feature of vent and seep environments.

Although this trend in increasing generic diversity among bivalves is roughly mirrored by an increase in the number of seep-bearing rock units (Fig 1), this pattern does not hold when seen in detail: (i) there is an increase in bivalve diversity from the Early to the Late Cretaceous despite a >50% decrease in the number of seep-bearing rock units; (ii) there are roughly identical numbers of seep-bearing rock units in the Early Cretaceous and in the Paleogene, but almost twice as many bivalve genera in the Paleogene; (iii) the number of seep-bearing rock units doubles from the Paleogene to the Neogene, accompanied by only a minor increase in bivalve diversity. Thus, we are confident that the observed pattern in bivalve diversity at seeps represents a real phenomenon, rather than being a sampling bias, although it is clear that the Paleozoic and early Mesozoic are still undersampled and likely contained higher numbers of bivalves at seeps.

Seep-dwelling dimerelloids are of low diversity during the Paleozoic, show a slight increase during the Jurassic and disappear after the Early Cretaceous (Fig 1). This pattern does not mirror the general Phanerozoic brachiopod diversity pattern of Paleozoic dominance, end-Permian decline, and low post-Paleozoic diversity [2]. Two observations indicate that this pattern is not significantly affected by sampling biases: first, despite the large number of seep-bearing rock units in the Early Cretaceous, there is only a single dimerelloid genus at seeps in this epoch. Second, since the first review of dimerelloid genera as potential seep-inhabiting brachiopods in 1995 [23], only a single new dimerelloid genus has been described: the Carboniferous *Ibergirhynchia* [72]. During the same time interval, 18 new genera of seep-inhabiting bivalves have been described, including nine from the Mesozoic and Paleozoic (indicated by asterisks in Table 1). This indicates that despite being undersampled, the relative proportions of brachiopod and bivalve genera at Paleozoic and early Mesozoic seeps shown in Fig 1 are fairly robust. The diversity pattern also does not confirm the paradigm that vents and seeps were dominated by brachiopods during the Paleozoic and most of the Mesozoic and that chemosymbiotic bivalves took over only in the Late Cretaceous [4]. Instead, dimerelloid brachiopods and chemosymbiotic bivalves have coexisted at seeps for nearly half of the Phanerozoic (Late Devonian to Late Cretaceous, ~240 million years [19]). This raises the question whether chemosymbiotic bivalves have indeed "exploited these habitats better than brachiopods" ([4] Campbell and Bottjer, 1995, p. 323).

## Ecology of seep-inhabiting brachiopods

At modern seeps, coexisting taxa tend to be spatially separated because different organisms require different types and amounts of geofuels, and the distribution of these geofuels is in turn controlled by flow rates and the resulting geochemical gradients [31, 159]. For example, among two species of vesicomyid clams at seeps in Monterey Canyon, *Archivesica kilmeri* requires 10 times higher ambient sulfide concentrations than *Calyptogena pacifica*, and consequently *C. pacifica* occupies the periphery of the seep where sulfide flux is low, whereas *A. kilmeri* lives in the sulfide-rich center of the seep [160]. Analogous faunal distribution patterns in relation to geochemical gradients can be traced into the fossil record: mollusks at Cretaceous seeps show similar zonation as their modern analogs [108, 109], and predation scar frequencies in Oligocene chemosymbiotic bivalves are inversely related to the different, assumed sulfide requirements of these species, most likely because the more sulfidic areas were avoided by predators and hence the bivalves with the highest sulfide requirements were spared from predation [161].

The Cretaceous seep-inhabiting dimerelloid brachiopod *Peregrinella* provides a particularly intriguing case of a geochemically controlled distribution pattern: *Peregrinella* was shown to have grown to much larger size at seeps with slow, diffusive fluid flow compared to sites with strong, advective fluid flow [38]. Because advective fluid flow releases more sulfide to the seabed than diffusive flow [146, 159], this pattern was interpreted as evidence that sulfide-rich seep sites were not ideal for *Peregrinella* and that bacterial, aerobic methane oxidation might have played a more prominent role in its nutrition [38]. That study used the abundance of early diagenetic fibrous cement in the seep limestone as a proxy for seepage intensity–with cement abundance positively correlated with seepage intensity [135]–and the authors pointed out that various other dimerelloids, including the very large *Dzieduszyckia*, lived at sites with very abundant seep cement (*Anarhynchia* even at an ancient hydrothermal vent site), and concluded that different dimerelloids might have had different feeding strategies [38].

Contrary to this claim, here we argue that seep-inhabiting dimerelloids in general relied on hydrocarbon-oxidizing bacteria for nutrition, rather than on sulfide oxidation. The presence

of methane and oil in the water column results in rapid growth of bacterioplankton that takes advantage of these energy sources [162, 163]. We put forward the hypothesis that dimerelloids thrived by feeding on the abundant bacterioplankton at seeps where high amounts of hydrocarbon geofuels effused into bottom waters. To the best of our knowledge, there is no present-day example of a species at seeps with this feeding strategy. The closest modern analogs are probably certain species of stalked barnacles (Cirripedia) living at vents in the West Pacific Ocean [164] and near the Antarctic Peninsula [165], which are adapted to feeding on very fine particles such as bacteria and fine debris [164]. In the following, we go through all pre-Cretaceous (that is: pre-*Peregrinella*) instances of dimerelloids at hydrocarbon seeps to outline our arguments for (i) fluid composition and flow intensity at each site, and (ii) their implications for the dimerelloids' preference for hydrocarbons over sulfide.

*Cooperrhynchia.* The Late Jurassic dimerelloid *Cooperrhynchia* is known from a single deposit only, were it is not superabundant but instead occurs in patches ([71] Sandy and Campbell, 1994; SK, own observation). The most common chemosymbiotic bivalve at this site is a solemyid [67], a group known to tolerate only low sulfide concentrations [159]. Similarly, the scarcity of $^{13}$C-depleted crocetane and the presence of $^{13}$C-depleted biphytane in the deposit with *Cooperrhynchia* [166] is typical of seep limestones that resulted from diffusive seepage [135], which would have come along with low sulfide concentrations close to the seabed.

*Anarhynchia.* This is the only dimerelloid genus yet known from both seeps and vents. An Early Jurassic seep deposit in northern British Columbia is dominated by *Anarhynchia smithi* and contains virtually no other fossils [18]. Despite the presence of early diagenetic fibrous cement, *Anarhynchia smithi* probably lived in a low-sulfide environment. It occurred during a geologic time interval known for its particularly low seawater sulfate concentration [167], which most likely resulted in reduced sulfide availability at seeps (cf. [9] Kiel, 2015; [145] Wortmann and Paytan, 2012) and hence also increased methane availability. Also of Early Jurassic age is a hydrothermal vent deposit in the Franciscan Complex in California, USA, at which *Anarhynchia* cf. *gabbi* is quite common [16, 168]. This occurrence at a hydrothermal vent site undoubtedly indicates that *Anarhynchia* was able to live at or near a strong sulfide source. But this does not necessarily contradict our hypothesis: hydrothermal vents are known to emit considerable amounts of methane, to the extent that for example the Rainbow, Snake Pit, and Logatchev vent sites on the Mid-Atlantic Ridge are inhabited by *Bathymodiolus* species hosting both thiotrophic and methanotrophic symbionts [169, 170].

*Sulcirostra.* Also of Early Jurassic age are seep deposits with *Sulcirostra* in eastern Oregon, USA; these are monospecific mass occurrences of *Sulcirostra paronai* that apparently lack bivalves and other fossils [17]. Analogously to the reasoning for the seep-inhabiting *Anarhynchia* above that lived in a low-sulfate ocean, we consider these occurrences low-sulfide environments. The great abundance of early diagenetic fibrous cement on the other hand is in accord with advective seepage, which is in favor of high sulfide production; but such production was necessarily still limited by the sulfate concentration of pore waters. Maybe even more interestingly, the *Sulcirostra* deposit contains pyrobitumen and its authigenic carbonate phases are only moderately $^{13}$C-depleted ($\delta^{13}$C values as low as –23.5‰), both agreeing with oil seepage [17].

*Halorella.* An argument for a preference for low-sulfide, diffusive seeps with abundant hydrocarbons in the bottom water analogous to that for *Peregrinella* can be made for the Triassic dimerelloid *Halorella*. In seep deposits in Oregon, *Halorella* occurs in rock-forming quantities, reaches almost 10 cm in size, and chemosymbiotic bivalves are rare or absent [15]. In contrast, in seep deposits in Turkey, *Halorella* is rare to common but never abundant, it never exceeds 45 mm in size, and it co-occurs with abundant inferred chemosymbiotic bivalves,

including two species of Kalenteridae and one anomalodesmatan [21, 22]. Assuming that the abundant inferred chemosymbiotic bivalves relied on thiotrophy rather than methanotrophy, this indicates a stronger sulfide flux at the seeps with abundant bivalves compared to those without. Consequently, also *Halorella* appears to have preferred seeps with less sulfide and more methane or other hydrocarbons.

***Ibergirhynchia.*** Early Carboniferous limestones with a mass occurrence of the dimerelloid *Ibergirhynchia* on top of a drowned atoll reef probably represent the most unusual Phanerozoic seep deposit reported to date [14]. Oil–as indicated by the presence of abundant pyrobitumen in the reef and seep limestones–passed through fissures of the Devonian atoll reef and fueled a chemosynthesis-based community on top of the reef. Migration of abundant oil through the Iberg reef apparently occurred in the latest early Carboniferous when the potential source rock, the Middle Devonian Wissenbach black shale, was in the oil window [171]. Due to the lack of a sedimentary cover, a large amount of the emitted geofuels necessarily entered the bottom water and consequently favored bacterioplankton growth, which, in turn, would have been suitable for the filter-feeding brachiopods.

***Dzieduszyckia.*** The Moroccan deposit with *Dzieduszyckia* contains abundant early diagenetic fibrous cement [13]. If the seepage fluids had been dominated by methane, such a pattern would suggest a sulfide-rich environment; a context similar to that of the *Sulcirostra* deposits of eastern Oregon. However, the presence of pyrobitumen and trace metal patterns reveal that the Upper Devonian limestone with *Dzieduszyckia* represents an oil-seep deposit [13, 144]. At oil seeps, where both oil and accessory methane escape the seabed, these geofuels facilitate bacterioplankton growth [162, 163], resulting in conditions favorable for the colonization by dimerelloid brachiopods. The Middle Devonian Hollard Mound seep deposit is also typified by abundant early diagenetic fibrous cement [132], but contains a mass occurrence of modiomorphid bivalves (*Ataviaconcha*) instead of dimerelloids [19]. Unlike the *Dzieduszyckia* oil-seep deposit, thermogenic or abiogenic methane, deriving from the underlying volcaniclastics, have been inferred as dominant geofuels of the Hollard Mound seep [172, 173]. The patterns found for the Devonian seep deposits consequently agree with the hypothesized resource partitioning between hydrocarbon-dependent brachiopods and sulfide-dependent bivalves.

In summary, there are several lines of evidence suggesting that, unlike most bivalves, dimerelloid brachiopods at Paleozoic and Mesozoic seeps were dependent on hydrocarbon rather than sulfide oxidation. Although we have made a strong case for filter-feeding on bacterioplankton for dimerelloid brachiopods, we cannot exclude the possibility that dimerelloids hosted episymbiotic bacteria on the surface of the lophophor instead of feeding on bacterioplankton. However, we do not consider this further because (i) such adaptation is unknown from living brachiopods, (ii) it would be very difficult to proof based on fossil evidence, and (iii) it does not change or add much to our hypothesis. Like episymbiosis, endosymbiosis cannot be fully excluded either. A few animals with symbionts oxidizing short-chain alkanes are known [174]. Yet, because of the lack of features in the brachiopod bauplan that are essential for endosymbiosis in other groups of animals, we consider it unlikely that the seep-dwelling dimerelloids harbored chemosymbiotic bacteria in their soft tissue.

Perhaps contrary to the scenario proposed here might be the lack of brachiopod-dominated seeps during the mid-Cretaceous to early Eocene period of low marine sulfate concentrations [9, 145]. If our scenario is correct, this time interval should have been favorable for dimerelloid brachiopods at seeps. The only explanation we can offer is that dimerelloids went extinct in the Barremian with the disappearance of *Peregrinella* [38], so that simply no suitable brachiopods were around to take advantage of the methane-rich seeps. This hypothesis is based on the following lines of evidence:

i.  the inclusion of the Cretaceous to present-day Cryptoporidae in the dimerelloids is questionable, so that *Peregrinella* is probably indeed the geologically youngest dimerelloid [175];

ii.  save for the Silurian *Septatrypa*, only dimerelloids have been able to dominate fossil seep sites, indicating that they possessed some pre-adaptation to successfully invade this habitat;

iii.  although other brachiopods, namely various terebratulids, have been found at fossil seeps [28, 33, 34, 37], they never formed mass occurrences like dimerelloids, and hence did not fill the same ecologic niche as dimerelloids;

iv.  the stratigraphic ranges of seep-inhabiting dimerelloids rarely overlap; this is particularly obvious for the three very large-sized genera *Dzieduszyckia*, *Halorella*, and *Peregrinella*, which are considered phylogenetically closely related ([28] Sandy, 2010, fig 9.6 therein) but are separated stratigraphically by 80 to 130 million years. This suggests that the genera discussed above represent repeated and temporarily very successful radiations into seep environments, which must be derived from as-yet unknown 'ghost dimerelloids' that may have been small and may have lived in cryptic or erosional settings (as suggested earlier for dimerelloids, cf. [176] Ager 1965).

Thus, the apparently only brachiopod lineage with the ability (or a trait) to colonize and to become a dominant member of vent and seep communities became extinct during the Early Cretaceous. This could explain why no brachiopod mass occurrences have been found at seeps during the theoretically favorable 'low sulfate interval' in the mid-Cretaceous to early Eocene. Furthermore, this also argues against the possibility that in the Cenozoic brachiopods were outcompeted at seeps by epifaunal bivalves or by bivalves with methanotrophic symbionts.

An analogous case of partitioning of resources instead of competition for them was recently made for Phanerozoic shallow-water brachiopods and bivalves in general [3]. This allows us to put forward the following scenario: resource partitioning controlled the evolutionary relationship between brachiopods and bivalves both in shallow marine habitats as well as at deep-water hydrocarbon seeps. But in seep environments, the animals were partitioning resources whose availability was controlled by fluid composition and flow intensity rather than by photosynthetic primary production, and hence the Phanerozoic diversity pattern of seep-dwelling animals differs from that of their shallow water relatives.

## Conclusions

The diversity patterns of brachiopods and chemosymbiotic bivalves at seeps through the Phanerozoic indicate an interesting combination of evolutionary trajectories. The diversity of infaunal chemosymbiotic bivalves at seeps mirrors their diversity in shallow-marine environments, whereas epifaunal and semi-infaunal chemosymbiotic bivalves are unique to vent and seep ecosystems and are not found in shallow water. Brachiopod diversity at seeps is unlike the global shallow-marine trend, is unrelated to the diversity of seep-dwelling bivalves, and instead indicates long-term coexistence of the two clades. Therefore, bivalves and brachiopods have probably not been competing for the same resources but instead partitioned the food sources resulting from the two most common categories of geofuels in seepage fluids: (i) hydrogen sulfide and (ii) methane and oil-derived components. Chemosymbiotic bivalves mostly relied on sulfide-oxidizing symbionts for nutrition, for the brachiopods bacterial aerobic oxidation of methane and of other hydrocarbons played a more prominent role. The distribution and availability of hydrogen sulfide and methane at seeps is governed by geochemical gradients and ocean chemistry, which in turn should ultimately have controlled whether bivalves or brachiopods dominated hydrocarbon seeps, both in space and through geologic time.

## Acknowledgments

We thank Krzysztof Hryniewicz (Warsaw) and three anonymous reviewers for their critical reading of the manuscript and its earlier versions.

## Author Contributions

**Conceptualization:** Steffen Kiel, Jörn Peckmann.

**Data curation:** Steffen Kiel, Jörn Peckmann.

**Formal analysis:** Steffen Kiel, Jörn Peckmann.

**Investigation:** Steffen Kiel, Jörn Peckmann.

**Methodology:** Steffen Kiel, Jörn Peckmann.

**Writing – original draft:** Steffen Kiel, Jörn Peckmann.

**Writing – review & editing:** Steffen Kiel, Jörn Peckmann.

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
