## [Decision Letter · Decision Letter 0]

7 Aug 2019

PONE-D-19-16788

Resource partitioning among brachiopods and bivalves at ancient hydrocarbon seeps: a hypothesis

PLOS ONE

Dear Prof. Peckmann,

Thank you for submitting your manuscript to PLOS ONE. After careful consideration, we feel that it has merit but does not fully meet PLOS ONE’s publication criteria as it currently stands. Therefore, we invite you to submit a revised version of the manuscript that addresses the points raised during the review process.

We would appreciate receiving your revised manuscript by Sep 21 2019 11:59PM. To enhance the reproducibility of your results, we recommend that if applicable you deposit your laboratory protocols in protocols.io, where a protocol can be assigned its own identifier (DOI) such that it can be cited independently in the future. For instructions see: http://journals.plos.org/plosone/s/submission-guidelines#loc-laboratory-protocols

We look forward to receiving your revised manuscript.

Kind regards,

Jürgen Kriwet

Academic Editor

PLOS ONE

Journal Requirements:

Additional Editor Comments (if provided):

Dear authors,

This manuscript represents a very interesting study that has a lot of merits. Nevertheless, I would like to ask you to carefully address alll comments of the two reviewers.

Best wishes,

Jürgen

Reviewers' comments:

Reviewer's Responses to Questions

**Comments to the Author**

1. Is the manuscript technically sound, and do the data support the conclusions?

Reviewer #1: Yes

Reviewer #2: Yes

2. Has the statistical analysis been performed appropriately and rigorously? 

Reviewer #1: Yes

Reviewer #2: Yes

3. Have the authors made all data underlying the findings in their manuscript fully available?

Reviewer #1: Yes

Reviewer #2: Yes

4. Is the manuscript presented in an intelligible fashion and written in standard English?

Reviewer #1: Yes

Reviewer #2: Yes

5. Review Comments to the Author

Reviewer #1: This is a review of manuscript „Resource partitioning among brachiopods and bivalves at ancient hydrocarbon seeps: a hypothesis” by Steffen Kiel and Joern Peckmann. The manuscript tackles and interesting topic of seemingly contrasting brachiopod and bivalve evolutionary histories at seeps throughout the Phanerozoic. The authors present an interesting hypothesis that bivalves and brachiopods utilized different resources, and that the contrasting evolutionary pathways are apparent rather than real. Authors present evidence to support this hypothesis, which makes the manuscript a quality work eligible for publication after a minor revision.

I have reviewed the previous version of this manuscript, where I have pointed several drawbacks in the reasoning, which were improved since then. The manuscript is properly referenced, although there are several important references missing, as I outline below. There are some minor drawbacks left to be corrected, outlined below.:

Line 40: please don’t get overly enthusiastic, it is difficult to judge what is most, and what is least extreme environment, it all depends on for whom; “…a similar pattern was also seen at deep-sea hydrothermal vents and hydrocarbon seeps” is enough

Line 182: we are unable to say whether they indeed have never developed methanotrophic symbioses, or they developed them only to have those symbioses extinct; better to write: Remarkable in this context is that other bivalve families with intracellular symbionts have not developed methanotrophic symbioses, or have not developed methanotrophic symbioses that would survive to the present day

Line 261-263: Phreagena kilmeri, rather than Archivesica kilmeri (check out vesicomyid taxonomy update paper by Krylova and Sahling (2010), PLoS one; line 262: C. pacifica, rather than P. pacifica

Line 308-309: Remove the sentence “Despite the presence of early diagenetic fibrous cement, Anarhynchia smithi probably lived in a low sulfide environment.” It doesn’t bring anything into the discussion, later you give some reasons why this might have been a low sulfide seep.

Line 332-333: There are two Greylock Butte seep deposits; one contains micrite, mass accumulations of Halorella up to 10 cm along longer axis, and no other fossils; the second Greylock Butte deposit contains somewhat rare Halorella up to 3-4 cm along longer axis, and some molluscs (anomalodesmatan and modiomorphid bivalves, among others); it is not true that both seep deposits in Oregon contain mass accumulations of Halorella, both are different, with second one being somewhat similar to Turkish deposit; please rewrite accordingly.

Line 350-351: remove “which, in turn, was suitable for the filter-feeding brachiopods”. Unnecessary.

Line 385-386: Eucalathis methanophila at Omagari occurs in some sort of monospecific brachiopod accumulations, although it doesn’t for such mass accumulations as dimerelloids did, please consider that; please add citation of Hryniewicz et al. (2019), describing another non-dimerelloid brachiopod from seep, Neoliothyrina nakremi

Another figure, with two schematic subfigures illustrating geochemical gradients and fauna at low-sulfide seeps with dimerelloids, and high sulfide seeps with bivalves, respectively, would be useful in this manuscript.

Table 1: clarify why you asterisk fossil genera erected after 1995 (presumably because after Campbell and Bottjer paper); you should asterisk also extant genera with fossil record erected after 1995 (they were unknown for Campbell and Bottjer, just as the fossil genera were)

Neogene:

Cubatea, rather than Cubathea, consider asterisking Elliptiolucina, Meganodontia according to my previous comments

Adulomya vs. Pleurophopsis: in your (SK) recent paper on Japanese Neogene vesis you consider Adulomya and Pleurophopsis as synonyms, with Pleurophopsis having priority. These seems to be true at least for Neogene species. Please be consistent here.

Samiolus from your 2017 paper (SK) paper on Miocene chemosymbiotic seep bivalves from Italy is missing – why? Also “Anodontia” from Ca’Fornace is missing (citation: Kiel et al. 2018), and Megaxinus from Stirone river (citation: Kiel and Taviani 2018), please complement the reference list and adjust fig. 1 according to missing genera

Paleogene:

Cubatea, not Cubathea

add Rhacothyas (citation: Hryniewicz et al. 2019), also Solemya species recorded there; adjust figure 1

Adulomya/Pleurophopsis problem again

Jurassic: Please note that there were 15 seep carbonates in Sassenfjorden area, with 3 of them Jurassic in age and the 8 Cretaceous (remainder not dated; details of stratigraphy in Wierzbowski et al. 2011, available from me on request). Therefore, putting all Sassenfjorden seep carbonates into Early Cretaceous category is an error. The significance of that is Solemya from one of the Jurassic seeps from Sassenfjorden, which is missing from your list and should be added (cite Hryniewicz et al. 2014, Zootaxa)

Table 2

be consistent and use “Fm” or “fm”; some of the terms used later are not lithostratigraphic in any way, but geographic (e.g. Beskidy Range, Musenalp, Wollaston Forland), therefore

Neogene: Ca’Fornace seep deposit missing (citation: Kiel et al. 2018)

Paleogene

Paleocene seep sites from Spitsbergen from Spitsbergen missing (citation: Hryniewicz et al. 2016, Palaeo3x, describing site, not fauna; do not cite 2019 paper here as it discusses the fauna)

Satsop Weatherwax seep deposit missing (citation: Hybertsen and Kiel, 2018, APP)

Late Cretaceous: separate Omagari lens and Yasukawa seep, these are two very different deposits

There are two Late Cretaceous seep deposits on Antarctic Peninsula; one on Seymour Island (Lopez de Bertodano Fm), one on Snow Hill Island (Snow Hill Island Formation), both are lumped into one here which is a mistake; worse, the deposit from Seymour Island is referred to as belonging to Snow Hill Island Formation, the seep-bearing lithologies of which do not even cropp out on Seymour Island; please correct that

Devonian

Regarding Sidi-Amar: in your 2007 paper you write about a limestone layer yielding Dzieduszyckia which was sampled, please include that next to “erratic limestone”; and “Devonian erratic limestones” does not look good as it suggest glacial erratics to most people, perhaps write “Devono-Carboniferous melange” or something like that

Despite numerous studies, I have seen the host lithological unit of Hollard Mound named just once, and that would be Pinacites limestone indeed (Peckmann et al. 1999); this is an informal unit, so small “l” and italics for Pinacites

Missing references:

Hryniewicz, K., Little, C.T.S., and Nakrem, H.A. 2014. Bivalves from the latest Jurassic–earliest Cretaceous hydrocarbon seep carbonates from Spitsbergen, Svalbard. Zootaxa 3859: 1–66.

Hryniewicz, K., Bitner, M.A., Durska, E., Hagström, J., Hjálmársdottir H.R., Jenkins, R.G., Little, C.T.S., Miyajima, Y., Nakrem, H.A., and Kaim, A. 2016. Paleocene methane seep and wood-fall marine environments from Spitsbergen, Svalbard. Palaeogeography, Palaeoclimatology,Palaeoecology 462: 41–56.

Hryniewicz, K., Amano, A., Bitner, M.A., Hagström, J., Kiel, S., Klompmaker, A.A., Mörs, T., Robins, C.M., and Kaim, A. 2019. A late Paleocene fauna from shallow-water chemosynthesis-based ecosystems, Spitsbergen, Svalbard. Acta Palaeontologica Polonica 64 (1): 101–141.

Hybertsen, F. and Kiel, S. 2018. A middle Eocene seep deposit with silicified fauna from the Humptulips Formation in western Washington State, USA. Acta Palaeontologica Polonica 63 (4): 751–768.

Kiel, S., Taviani, M. 2017. Chemosymbiotic bivalves from Miocene methane seep carbonates in Italy. Journal of Paleontology 91, 444–466.

Kiel, S. and Taviani, M. 2018. Chemosymbiotic bivalves from the late Pliocene Stirone River hydrocarbon seep complex in northern Italy. Acta Palaeontologica Polonica 63 (3): 557–568.

Kiel, S., Sami, M., and Taviani, M. 2018. A serpulid-Anodontia-dominated methane-seep deposit from the Miocene ofnorthern Italy. Acta Palaeontologica Polonica 63 (3): 569–577.

Krzysztof Hryniewicz

Reviewer #2: To authors

It is challenging paper which try to answer to one of the most interesting issue on faunal change in seep environment through the Phanerozoic; brachiopods vs bivalves in seeps.

Although your hypothesis, seep obligate dimerelloid brachiopods had mainly consumed methanotrophic and/or hydrocarbon-consuming bacteria, is seemed to established upon many assumptions, I think the idea based on you detailed review on the current our knowledge is fair. The paper would be a starting point to make better understanding of on brachiopods vs bivalves in seeps. Followings are minor point to be revised.

Minor points:

l. 72-76: Although I know authors stated in the discussion parts, it would be better to mention about non-dimerelloid brachiopod, e.g. terebratulide Eucalathis, occurrences in seep.

l. 238-240: Authors mentioned that the seep dwelling brachiopods disappeared after the Early Cretaceous, however, there are several occurrences of terebratulide brachiopods in Late Cretaceous and Cenozoic seeps. May be authors think those terebratulide brachiopods are not seep obligate group, author should state their thinking with reasonable reason before this sentence.

l. 262: not P. pacifica, C. pacifica

l. 287-290: So, do you think the dimerelloid brachiopods fed on free living hydrocarbon-consuming microbes only? Don’t you think they have epi-symbiotic bacteria on the surface of lophophor? Why you can exclude this possibility? Please state.

It is also better to argue or state the internal morphology of dimerelloid compared to other brachiopods. If they have special features, please state.

Table 1: I couldn’t understand why author put Conchocele bivalves as “semi-infauna.” I know they sometimes lived on the sea floor, like in Sea of Okhotsk, but in most cases they lived within sediments, beneath the sea floor. So, it shout be categorized into “infauna”.

l. 374-379: It is just comment. As authors already noted, it is big contrary that the dimerelloids couldn’t flourished in low sulfate concentration period, L. Cretaceous.

l. 778: delete “pdf ed.”

6. PLOS authors have the option to publish the peer review history of their article (what does this mean?). If published, this will include your full peer review and any attached files.

Reviewer #1: Yes: Krzysztof Hryniewicz

Reviewer #2: No

---

## [Author Response · Author response to Decision Letter 0]

14 Aug 2019

Note that the color code of the response letter will not show in this box.

PONE-D-19-16788

Resource partitioning among brachiopods and bivalves at ancient hydrocarbon seeps: a hypothesis

Comments to the Author

Reviewer #1: This is a review of manuscript „Resource partitioning among brachiopods and bivalves at ancient hydrocarbon seeps: a hypothesis” by Steffen Kiel and Joern Peckmann. The manuscript tackles and interesting topic of seemingly contrasting brachiopod and bivalve evolutionary histories at seeps throughout the Phanerozoic. The authors present an interesting hypothesis that bivalves and brachiopods utilized different resources, and that the contrasting evolutionary pathways are apparent rather than real. Authors present evidence to support this hypothesis, which makes the manuscript a quality work eligible for publication after a minor revision.

I have reviewed the previous version of this manuscript, where I have pointed several drawbacks in the reasoning, which were improved since then. The manuscript is properly referenced, although there are several important references missing, as I outline below. There are some minor drawbacks left to be corrected, outlined below.:

Line 40: please don’t get overly enthusiastic, it is difficult to judge what is most, and what is least extreme environment, it all depends on for whom; “…a similar pattern was also seen at deep-sea hydrothermal vents and hydrocarbon seeps” is enough

Reply: Point taken; we have changed wording accordingly.

Line 182: we are unable to say whether they indeed have never developed methanotrophic symbioses, or they developed them only to have those symbioses extinct; better to write: Remarkable in this context is that other bivalve families with intracellular symbionts have not developed methanotrophic symbioses, or have not developed methanotrophic symbioses that would survive to the present day

Reply: A fair comment, but also a bit of a nit-picking comment. Since there is no point in discussing this is detail, we have changed “other families with intracellular symbionts have never developed methanotrophic symbiosis” to “other families with intracellular symbionts have apparently not developed methanotrophic symbiosis “.

Line 261-263: Phreagena kilmeri, rather than Archivesica kilmeri (check out vesicomyid taxonomy update paper by Krylova and Sahling (2010), PLoS one; line 262: C. pacifica, rather than P. pacifica

Reply: The species assigned by Krylova and co-workers to Phreagena have not been retrieved as a monophyletic group in any molecular analysis of the vesicomyids; Archivesica is used here in the sense of Kiel (2016, Proc. Royal Society B 283: 2016233). Corrected ‘P. pacifica’

Line 308-309: Remove the sentence “Despite the presence of early diagenetic fibrous cement, Anarhynchia smithi probably lived in a low sulfide environment.” It doesn’t bring anything into the discussion, later you give some reasons why this might have been a low sulfide seep.

Reply: We disagree. Omitting this information may come along with the impression that we have ignored evidence that is not in favor of our interpretation. Although the evidence detailed here (abundant fibrous cement) may argue for a high sulfide environment, there is other evidence – and we think this is stronger evidence – in favor of a low sulfide environment, which is detailed in the main text. For the sake of good scientific practice, we prefer to keep on mentioning evidence not in favor of our overall interpretation.

Line 332-333: There are two Greylock Butte seep deposits; one contains micrite, mass accumulations of Halorella up to 10 cm along longer axis, and no other fossils; the second Greylock Butte deposit contains somewhat rare Halorella up to 3-4 cm along longer axis, and some molluscs (anomalodesmatan and modiomorphid bivalves, among others); it is not true that both seep deposits in Oregon contain mass accumulations of Halorella, both are different, with second one being somewhat similar to Turkish deposit; please rewrite accordingly.

Reply: The differences between the two Graylock Butte outcrops as outlined by the reviewer may indeed exists, however, based on our own field experience we cannot exclude that the Graylock Butte 1 and 2 deposits are only one deposit in reality. The large deposits are only approx. 200 m apart in outcrop, but may be connected in the subsurface. Bivalves were only found in the Graylock Butte 2 deposit by us, but only at one meter-scale spot at its margin (Peckmann et al. 2011). We cannot exclude that similar accessory fossil assemblages occur within the large Graylock Butte 1 deposit (70 m in width) that are not exposed in outcrop. It is certainly NOT correct, however, that the second Graylock Butte deposit is more similar to some of the Turkish deposits (described in Kiel et al. 2017) than to the first Graylock Butte deposit. Therefore, we prefer not to make a case for facies differentiation among the Graylock Butte deposits. Anyway, even if we would make such a case, it would be in favor of our hypothesis: large brachiopods and lack of bivalves coincides with high methane/low sulfide for Graylock Butte 1, whereas somewhat smaller (?) but certainly fewer brachiopods and the very few bivalves coincide with less methane/higher sulfide flux for Graylock Butte 2.

Line 350-351: remove “which, in turn, was suitable for the filter-feeding brachiopods”. Unnecessary.

Reply: We feel that this clarification is of use. The reviewer is a leading expert of the research topic, but not every reader will be equally familiar with the topic. Anyway, we toned down wording to avoid the impression that we use this statement as an argument and not for clarification. This, indeed, would be circular reasoning. We have changed “was suitable” to “would have been suitable”.

Line 385-386: Eucalathis methanophila at Omagari occurs in some sort of monospecific brachiopod accumulations, although it doesn’t for such mass accumulations as dimerelloids did, please consider that; please add citation of Hryniewicz et al. (2019), describing another non-dimerelloid brachiopod from seep, Neoliothyrina nakremi

Reply: Eucalathis methanophila occurs in small numbers among a mass accumulation of small and diverse gastropods at the Omagari site. This does not resemble in any way the mass accumulations of dimerelloids – and should never have been called a ‘monospecific assemblage’ in the title of that paper in the first place. The citation was added.

Another figure, with two schematic subfigures illustrating geochemical gradients and fauna at low-sulfide seeps with dimerelloids, and high sulfide seeps with bivalves, respectively, would be useful in this manuscript.

Reply: We see the point and had thought about such a figure ourselves. In the end, we had opted against such schematic figure because there are not two simple end-member systems; too many factors impact the modes of seepage at the different sites discussed in the manuscript. We are afraid that we are not yet at a point, where we are able to include all steering mechanisms in a simple sketch.

Table 1: clarify why you asterisk fossil genera erected after 1995 (presumably because after Campbell and Bottjer paper); you should asterisk also extant genera with fossil record erected after 1995 (they were unknown for Campbell and Bottjer, just as the fossil genera were)

Reply: This was indeed insufficiently explained. In the discussion section we compare the number of newly described seep brachiopods genera since Mike Sandy’s (1995) review to the number of new seep-inhabiting bivalve genera described since the same year. We made this clearer in the discussion (lines 244 to 247 in the revised ms) and in the table caption. Per suggestion of the reviewer, we now also mark the new genera based on living species (Elliptiolucina, Meganodontia etc.), as well as the one new dimerelloid genus since 1995 (Ibergirhynchia). We have also adjusted the numbers in the text (line 246 in the revised ms).

Neogene:

Cubatea, rather than Cubathea, consider asterisking Elliptiolucina, Meganodontia according to my previous comments

Reply: Thanks for spotting the typo in Cubatea; see also previous comment.

Adulomya vs. Pleurophopsis: in your (SK) recent paper on Japanese Neogene vesis you consider Adulomya and Pleurophopsis as synonyms, with Pleurophopsis having priority. These seems to be true at least for Neogene species. Please be consistent here.

Reply: Thanks for spotting this; we removed Adulomya from the list.

Samiolus from your 2017 paper (SK) paper on Miocene chemosymbiotic seep bivalves from Italy is missing – why? Also “Anodontia” from Ca’Fornace is missing (citation: Kiel et al. 2018), and Megaxinus from Stirone river (citation: Kiel and Taviani 2018), please complement the reference list and adjust fig. 1 according to missing genera

Reply: Samiolus was not considered as a bathymodiolin by Kiel & Taviani, just as a mytilid. Hence it does not qualify as ‘chemosymbiotic bivalve’ and is consequently not included in this list. Anodontia and Megaxinus were added to the list.

Paleogene:

Cubatea, not Cubathea

add Rhacothyas (citation: Hryniewicz et al. 2019), also Solemya species recorded there; adjust figure 1

Reply: Rhacothyas was added to the list; Solemya was already in the list

Adulomya/Pleurophopsis problem again

Reply: Adulomya was removed.

Jurassic: Please note that there were 15 seep carbonates in Sassenfjorden area, with 3 of them Jurassic in age and the 8 Cretaceous (remainder not dated; details of stratigraphy in Wierzbowski et al. 2011, available from me on request). Therefore, putting all Sassenfjorden seep carbonates into Early Cretaceous category is an error. The significance of that is Solemya from one of the Jurassic seeps from Sassenfjorden, which is missing from your list and should be added (cite Hryniewicz et al. 2014, Zootaxa)

Reply: Sassenfjorden was added to the list of Jurassic sites/rock units and Solemya was added to the list of Jurassic infaunal bivalves. Thanks for correcting this.

Table 2

be consistent and use “Fm” or “fm”; some of the terms used later are not lithostratigraphic in any way, but geographic (e.g. Beskidy Range, Musenalp, Wollaston Forland), therefore

Reply: We replaced the one instance of “fm” by “Fm”. 

Neogene: Ca’Fornace seep deposit missing (citation: Kiel et al. 2018)

Reply: It is not the intention of this table to list all seep deposits, but just the seep-bearing rock units. This is clearly stated in the table caption. Furthermore, the Ca’Fornace limestone was found float and its exact provenance is unclear (Kiel et al. 2018). 

Paleogene

Paleocene seep sites from Spitsbergen from Spitsbergen missing (citation: Hryniewicz et al. 2016, Palaeo3x, describing site, not fauna; do not cite 2019 paper here as it discusses the fauna)

Reply: added

Satsop Weatherwax seep deposit missing (citation: Hybertsen and Kiel, 2018, APP)

Reply: As above, the seep-bearing rock unit (Humptulips Fm.) is already in the list. 

Late Cretaceous: separate Omagari lens and Yasukawa seep, these are two very different deposits

Reply: They are from the same rock unit; no point in separating them here.

There are two Late Cretaceous seep deposits on Antarctic Peninsula; one on Seymour Island (Lopez de Bertodano Fm), one on Snow Hill Island (Snow Hill Island Formation), both are lumped into one here which is a mistake; worse, the deposit from Seymour Island is referred to as belonging to Snow Hill Island Formation, the seep-bearing lithologies of which do not even cropp out on Seymour Island; please correct that

Reply: corrected and number on figure 1 adjusted; note that we also changed the word ‘Formations’ in both the legend and the caption of the X-axis to ‘rock units’. 

Devonian

Regarding Sidi-Amar: in your 2007 paper you write about a limestone layer yielding Dzieduszyckia which was sampled, please include that next to “erratic limestone”; and “Devonian erratic limestones” does not look good as it suggest glacial erratics to most people, perhaps write “Devono-Carboniferous melange” or something like that

Reply: changed to ‘Devonian-Carboniferous mélange’

Despite numerous studies, I have seen the host lithological unit of Hollard Mound named just once, and that would be Pinacites limestone indeed (Peckmann et al. 1999); this is an informal unit, so small “l” and italics for Pinacites

Reply: accepted

Missing references:

Hryniewicz, K., Little, C.T.S., and Nakrem, H.A. 2014. Bivalves from the latest Jurassic–earliest Cretaceous hydrocarbon seep carbonates from Spitsbergen, Svalbard. Zootaxa 3859: 1–66.

Hryniewicz, K., Bitner, M.A., Durska, E., Hagström, J., Hjálmársdottir H.R., Jenkins, R.G., Little, C.T.S., Miyajima, Y., Nakrem, H.A., and Kaim, A. 2016. Paleocene methane seep and wood-fall marine environments from Spitsbergen, Svalbard. Palaeogeography, Palaeoclimatology,Palaeoecology 462: 41–56.

Hryniewicz, K., Amano, A., Bitner, M.A., Hagström, J., Kiel, S., Klompmaker, A.A., Mörs, T., Robins, C.M., and Kaim, A. 2019. A late Paleocene fauna from shallow-water chemosynthesis-based ecosystems, Spitsbergen, Svalbard. Acta Palaeontologica Polonica 64 (1): 101–141.

Hybertsen, F. and Kiel, S. 2018. A middle Eocene seep deposit with silicified fauna from the Humptulips Formation in western Washington State, USA. Acta Palaeontologica Polonica 63 (4): 751–768.

Kiel, S., Taviani, M. 2017. Chemosymbiotic bivalves from Miocene methane seep carbonates in Italy. Journal of Paleontology 91, 444–466.

Kiel, S. and Taviani, M. 2018. Chemosymbiotic bivalves from the late Pliocene Stirone River hydrocarbon seep complex in northern Italy. Acta Palaeontologica Polonica 63 (3): 557–568.

Kiel, S., Sami, M., and Taviani, M. 2018. A serpulid-Anodontia-dominated methane-seep deposit from the Miocene ofnorthern Italy. Acta Palaeontologica Polonica 63 (3): 569–577.

Reply: all appropriate references were added

Krzysztof Hryniewicz

Reviewer #2: It is challenging paper which try to answer to one of the most interesting issue on faunal change in seep environment through the Phanerozoic; brachiopods vs bivalves in seeps. Although your hypothesis, seep obligate dimerelloid brachiopods had mainly consumed methanotrophic and/or hydrocarbon-consuming bacteria, is seemed to established upon many assumptions, I think the idea based on you detailed review on the current our knowledge is fair. The paper would be a starting point to make better understanding of on brachiopods vs bivalves in seeps. Followings are minor point to be revised.

Minor points:

l. 72-76: Although I know authors stated in the discussion parts, it would be better to mention about non-dimerelloid brachiopod, e.g. terebratulide Eucalathis, occurrences in seep.

Reply: Point taken. We now refer to the publications on terebratulides at seeps. The following addition (bold letters) was made to the text:

Among brachiopods, only dimerelloid genera reported from geochemically confirmed seep deposits are included because (i) with a single exception (see below), only dimerelloids occurred at ancient seeps in rock-forming quantities; all other brachiopods reported from ancient seeps (including various terebratulids, i.e. [33-36]) represent minor faunal elements that most likely took advantage of exposed hard substrate [28], …

l. 238-240: Authors mentioned that the seep dwelling brachiopods disappeared after the Early Cretaceous, however, there are several occurrences of terebratulide brachiopods in Late Cretaceous and Cenozoic seeps. May be authors think those terebratulide brachiopods are not seep obligate group, author should state their thinking with reasonable reason before this sentence.

Reply: We have replaced “Seep-dwelling brachiopods” by “Seep-dwelling dimerelloids”.

l. 262: not P. pacifica, C. pacifica

Reply: corrected

l. 287-290: So, do you think the dimerelloid brachiopods fed on free living hydrocarbon-consuming microbes only? Don’t you think they have epi-symbiotic bacteria on the surface of lophophor? Why you can exclude this possibility? Please state.

Reply: A fair comment that is consistent with our hypothesis. We have added the following statement (lines 368 to 372): “ … we cannot exclude the possibility that dimerelloids hosted episymbiotic bacteria on the surface of the lophophor instead of feeding on bacterioplankton. However, we do not consider this further because (i) such adaptation is unknown from living brachiopods, (ii) it would be very difficult to proof based on fossil evidence, and (iii) it does not change or add much to our hypothesis.”

It is also better to argue or state the internal morphology of dimerelloid compared to other brachiopods. If they have special features, please state.

Reply: The first author (SK) has discussed this matter with several brachiopod specialists. Their view is that although many dimerelloids have rather long crurae (lophophor support structures), the length of the crurae does not allow any conclusions on the length of the lophophor itself. Hence we refrain from speculating about this matter.

Table 1: I couldn’t understand why author put Conchocele bivalves as “semi-infauna.” I know they sometimes lived on the sea floor, like in Sea of Okhotsk, but in most cases they lived within sediments, beneath the sea floor. So, it shout be categorized into “infauna”.

Reply: To our knowledge, the only actual observations on the mode of life of Conchocele come from the Sea of Okhotsk, where they were semi-infaunal. Hence we follow this view (as done in a previous assessment of epifaunal/infaunal trends among seep bivalves by Kiel 2015, Proc. Roy. Soc. B, 282: 20142908).

l. 374-379: It is just comment. As authors already noted, it is big contrary that the dimerelloids couldn’t flourished in low sulfate concentration period, L. Cretaceous.

Reply: This is correct. We do discuss this relationship in detail in the lines below (lines 380 to 394, submitted version; lines 383 to 397, revised version). 

l. 778: delete “pdf ed.”

Reply: We thank the reviewer for spotting this typo. It has been corrected.

---

## [Editor Report · Decision Letter 1]

19 Aug 2019

Resource partitioning among brachiopods and bivalves at ancient hydrocarbon seeps: a hypothesis

PONE-D-19-16788R1

Dear Dr. Peckmann,

We are pleased to inform you that your manuscript has been judged scientifically suitable for publication and will be formally accepted for publication once it complies with all outstanding technical requirements.

With kind regards,

Jürgen Kriwet

Academic Editor

PLOS ONE

Additional Editor Comments (optional):

Dear authors,

Thank you for considering all comments and suggestions by the reviewers and also to provide apmple arguments where you did not follow the reviewer's suggestions. I agreen with all your comments and arguments and thus consider your manuscript acceptal for publication in Plos One in its current form.

Kind regards,

Jürgen Kriwet
---

## [Editor Report · Acceptance letter]

27 Aug 2019

PONE-D-19-16788R1 

Resource partitioning among brachiopods and bivalves at ancient hydrocarbon seeps: a hypothesis 

Dear Dr. Peckmann:

I am pleased to inform you that your manuscript has been deemed suitable for publication in PLOS ONE. Congratulations! Your manuscript is now with our production department. 

With kind regards,

on behalf of

Dr. Jürgen Kriwet 

Academic Editor

PLOS ONE